# From the Laboratory to the Field: IMU-Based Shot and Pass Detection in Football Training and Game Scenarios Using Deep Learning

**DOI:** 10.3390/s21093071

**Published:** 2021-04-28

**Authors:** Maike Stoeve, Dominik Schuldhaus, Axel Gamp, Constantin Zwick, Bjoern M. Eskofier

**Affiliations:** 1Machine Learning and Data Analytics Lab, Department of Artificial Intelligence in Biomedical Engineering (AIBE), Friedrich-Alexander-Universität Erlangen-Nürnberg (FAU), 91052 Erlangen, Germany; bjoern.eskofier@fau.de; 2Adidas AG, 91074 Herzogenaurach, Germany; dominik.schuldhaus@adidas.com (D.S.); axel.gamp@adidas.com (A.G.); constantin.zwick@adidas.com (C.Z.)

**Keywords:** data analysis, activity recognition, sensor-signal-based machine learning, deep learning, wearable sensors, sport

## Abstract

The applicability of sensor-based human activity recognition in sports has been repeatedly shown for laboratory settings. However, the transferability to real-world scenarios cannot be granted due to limitations on data and evaluation methods. On the example of football shot and pass detection against a null class we explore the influence of those factors for real-world event classification in field sports. For this purpose we compare the performance of an established Support Vector Machine (SVM) for laboratory settings from literature to the performance in three evaluation scenarios gradually evolving from laboratory settings to real-world scenarios. In addition, three different types of neural networks, namely a convolutional neural net (CNN), a long short term memory net (LSTM) and a convolutional LSTM (convLSTM) are compared. Results indicate that the SVM is not able to reliably solve the investigated three-class problem. In contrast, all deep learning models reach high classification scores showing the general feasibility of event detection in real-world sports scenarios using deep learning. The maximum performance with a weighted f1-score of 0.93 was reported by the CNN. The study provides valuable insights for sports assessment under practically relevant conditions. In particular, it shows that (1) the discriminative power of established features needs to be reevaluated when real-world conditions are assessed, (2) the selection of an appropriate dataset and evaluation method are both required to evaluate real-world applicability and (3) deep learning-based methods yield promising results for real-world HAR in sports despite high variations in the execution of activities.

## 1. Introduction

In order to surpass oneself, performance monitoring has become a crucial part of training for athletes of all levels. As more and more commercial devices are available, wearable trackers based on inertial measurement units (IMUs) are increasingly used for the assessment of sport-specific movements of recreational athletes in team sports [1,2]. Examples are IMU-based systems such as Babolat Play or Qlipp tennis sensor detecting stroke and spin types and racket speed in tennis or the Zepp Play Soccer assessing kicks and sprints in football. However, in literature automatic activity recognition algorithms are commonly developed and evaluated in laboratory settings lacking proof of transferability to real-world scenarios [3,4,5]. Thereby, limitations on two different levels of the processing pipeline, data acquisition and evaluation, can be observed.

In data acquisitions in laboratory settings, several factors can contribute to restricted variation in the athletes movements. This implies less representative data and limited generalizability of the results to real-world scenarios. The sport-specific movements of interest are usually performed isolated or combined in predefined exercises. Due to fixed distances or even positions, the intensities of the movements are restricted. Often, a special technique or execution is set further limiting variations in the movements. The restriction of limited variation is intensified by the commonly low number of participants resulting from the tedious and labour-intensive data recording and labelling process [6]. Consequently, study populations are rather homogeneous. As every athlete has their own execution pattern for each activity defined by their weight, height, sex and strength, inter-subject variability can increase task complexity and might impact classification performance [7]. Recent studies showed that deep learning models outperform conventional machine learning methods as Support Vector Machines (SVMs) in sensor-based human activity recognition (HAR) in sports [8,9,10]. In contrast to conventional methods which require hand-crafted, domain-specific features, deep learning models automatically extract abstract features from sensor signals [11]. However, the training of neural networks generally requires large labelled datasets to achieve satisfactory performance explaining its rare use for sensor-based performance analysis in field-sports [6,12].

In the evaluation step, the algorithms are commonly tested on segmented samples of the sensor signal containing the target movements and preselected null class instances [7,13]. Thus, the influence of confounding movements is neglected, although they might produce similar sensor output as the sport-specific movements under investigation. In an uncontrolled, real-world setting a higher number of confounding, unanticipated movements can be expected. Besides, the distribution of events will change. Training or game data will include a higher proportion of null class samples corresponding to general activities as standing, walking, running or jumping. In response, effective methods are needed to identify those samples prior to classification.

There are several studies tackling individual aspects of the aforementioned limitations, for instance, by evaluating on competition data or using a moving window approach for evaluation to consider the final distribution of events [4,14]. However, to the best of our knowledge a holistic consideration of limitations is not investigated yet. Thus, in this work we aim to explore the potential of IMU-based activity recognition under real-world conditions considering both factors, the data acquisition and training step on the example of football, the most popular sport in many regions of the world [15]. In football, the number of shots and passes is an important discriminating factor between winning and loosing teams in the UEFA Champions League [16]. Previous work already showed the feasibility to classify full-instep kicks and side-foot kicks in laboratory settings applying a SVM classifier. However, the number of participants was limited to eleven male amateur players, data was recorded during controlled exercises, performance was evaluated on segmented samples and the null class was limited to dribbling instances [13].

Building on the findings of Schuldhaus et al. [13] our research contributes in the following ways: First, we evaluated the applicability of the reported features and SVM classifier for shot and pass detection versus a null class in real-world scenarios. Second, we analysed the potential of improving classification performance by using and comparing different deep learning architecture types. In contrast to previous approaches, we classified general shot and pass activities regardless of the kicking technique allowing more variations of movements. Hence, the classification task gains complexity but seems to be more relevant for match analysis. To cover variations in kick and pass movements and provide sufficient data for the development of deep learning algorithms, we acquired a large dataset containing laboratory as well as real world instances from training and competition of over 800 football players. The algorithms are compared in different scenarios gradually evolving from laboratory settings with various restrictions to the real-world case. Finer-grained ball contact labels are used to gain additional insights into confounding factors.

## 2. Related Work

In HAR external sensors are used to detect and analyse various human movements as, amongst others, walking, running, sleeping, cooking or driving [17,18,19,20]. Thereby application areas are manifold covering amongst others healthcare monitoring [21], sports performance evaluation [13,22], smart home [23,24] and autonomous driving [20,25]. Depending on the external sensor source, video-based and sensor-based HAR systems are distinguished. Due to their low price, light weight and small size IMUs have extensively been used in the field of sensor-based HAR in the past years [26]. For our study, we identified two relevant directions of research in the field of sensor-based HAR. First, HAR in field sports considering real-world applicability. Second, the development of deep learning models in HAR in sports. The following paragraphs shortly summarize recent achievements.

### 2.1. Towards Real-World Sensor-Based Activity Recognition in Field Sports

In general, a trend towards in-field use of wearable IMUs can be observed. After Camomilla et al. [27], already 62% of considered studies were conducted on training or simulated training data, 7% on competition data and only 28% in laboratory settings. However, included work was neither restricted to field sports, nor on activity recognition. Instead, assessment of motor capacity, technique analysis and physical demand was covered [27]. In contrast to the findings for IMU-based analysis in sports in general, sensor-based activity recognition in team sports mainly takes place in laboratory settings. However, publications considering real-world data for training and/or evaluation can be found and are summarized below.

McNamara et al. [14] detected bowling activities in cricket training and competition using a commercially available sensor comprising an IMU and global positioning system (GPS) fixed at the upper back of twelve highly skilled players. High sensitivity and specificity values of 99.0% and 74.0%, respectively, were reported for the competition scenario [14]. Similarly, Kelly et al. [28] integrate an accelerometer and GPS receiver between the shoulder blades of rugby players for the identification of tackles and collisions combining SVM and hidden conditional random field by AdaBoost. For three players, classification was tested on predefined, segmented match data achieving high recall (93%) and precision rates (96%). In [29], the bowling activity in cricket is analysed using various action profiles derived from IMU data. Conventional machine learning algorithms as SVM, k-nearest neighbor, Naïve Bayes and Random Forest (RF) as well as a feed-forward neural net were trained on statistical features extracted from the activity profiles to distinguish legal and illegal bowling actions. For the complex dataset, the RF achieved the best performance with an F-measure of 0.74. Again, testing was performed on segmented samples of the bowling motion [29].

In contrast to the studies listed above, several publication propose hierarchical systems that deal with the data distribution in real-world scenarios but use laboratory data for evaluation [3,4,5]. Rawashdeh et al. [3] detect overhead motions in a first step. Then they distinguish baseball throws from volleyball serves with an accuracy of 94.04% in the second step in order to count straining motions for shoulder and elbow overuse injury prevention [3]. In [4], different sensor combinations and locations are analysed for HAR in field hockey. The best set up with 4 sensors results in a classification accuracy of around 97%. Within the approach, a moving window is applied for evaluation purposes considering the final distribution of samples. However, data collection is restricted to specific field hockey activities of 11 players in a controlled environment. Another hierarchical approach was introduced by Nguyen et al. [5], who recorded nine different movements using IMUs and a pressure sensor at the feet of basketball players. In a first step they differentiate standing and moving activities. Afterwards time- and frequency domain features are extracted and basketball activities are classified by a SVM [5].

For the classification task tackled within our study, namely the recognition and analysis of shots and passes in football, several related studies were identified. Mitchell et al. [7] extract features from smartphone accelerometers using Discrete Wavelet Transform. They compare different classification methods on segmented samples of soccer and field-hockey activities extracted from game data. They propose a fusion of different classifiers achieving an average maximum F-measure of 87% [7]. Chawla et al. [30] used supervised machine learning to classify passes in soccer matches regarding their quality based on positional data with an accuracy of 90.2%. The identification of shots was not covered in their work [30]. Schuldhaus et al. [13] developed a hierarchical pipeline based on IMU data for the detection of full-instep and side-foot kicks. For full-instep kicks, a mean sensitivity of 95.6% is reported. In addition, the corresponding ball speed was estimated and highlight videos were generated. Data was recorded during controlled exercises and only considered dribbling as null class instances [13]. Kim and Kim [31] propose the computation of an impact measurement function from the acceleration signal and detect leg swings from angular velocity for the recognition of kicks. The evaluation was performed on 5 subjects during a kicking exercise. Confounding ball contacts were not part of the session [31]. In [32], the direction of penalty shots was classified from accelerometer data using traditional machine learning models as well as a convolutional neural net (CNN). The proposed CNN architecture outperformed the traditional methods reaching an accuracy of 53%. The dataset consisted of a real-life penalty shoot out of 4 players [32]. To the extent of our knowledge, there is no work investigating the sensor-based classification of different kick types under real-world conditions.

### 2.2. Deep Learning for Sensor-Based Human Activity Recognition in Sports

Recently, a shift towards the application of deep learning methods can be observed in many fields [33]. A major advantage thriving this development is that this subgroup of machine learning methods does not require time-consuming manual feature extraction [34]. Moreover, deep learning approaches outperform conventional machine learning methods in many fields, for example in vision-based HAR [6]. However, in the field of sensor-based HAR in sports, the application of deep learning methods is not as frequent as in other fields due to high data demands [6,35]. Kautz et al. [8] trained a CNN on activity recognition in beach volleyball and compared it to various other shallow classifiers in combination with generic feature extraction. Thereby, the CNN exceeds the accuracy of other classifiers by 16.0% [8]. CNNs are also applied for error classification in ski jumping. A comparison to a SVM and a Hidden Markov model showed superior performance, especially for noisy and biased sensor data [9]. Jiao et al. [10] adapt a vanillaCNN, VGG, Inception and ResNet architecture for golf swing classification and compared it to a SVM, showing that CNN based models can appropriately solve the classification task and achieve higher scores than the SVM. Besides CNNs, recurrent neural nets (RNNs) and long short-term memory nets (LSTMs) in particular are used to model long term dependencies. Rassem et al. [36] compare a CNN, LSTM and standard multilayer perceptron for the classification of cross-country skiing movements. The lowest classification error (1.6%) was reported for the LSTM [36]. For swing sports as tennis, badminton and golf, shot classification is a well known HAR problem. Anand et la. [37] compared a feature-based classifier, a CNN and a Bi-directional LSTM (BLSTM) network detecting and classifying shots in tennis, badminton and squash. Again, the deep learning-based models outperformed the shallow architecture. For tennis, the CNN resulted in a slightly higher f1-score of 93.8% compared to the LSTM, whereas for badminton and squash peak performance was reached by the BLSTM model achieving 78.9% and 94.6%, respectively [37]. In [38], a SVM with Radial Basis Function, a LSTM and a 2-Dimensional CNN are compared for forehand stroke classification in table tennis. Both deep learning models outperformed the SVM, whereas best performance with an f1-score of 99.02% was achieved by the LSTM model [38]. To the best of our knowledge, activity recognition in football was not address with deep learning-based models yet.

## 3. Materials and Methods

In the following section an overview of the proposed methods is provided. The data acquisition and the adaptation of the SVM from [13] is described. Consecutively, the development, implementation and optimization of the proposed deep-learning architectures, as well as the evaluation procedures are explained.

### 3.1. Dataset

The dataset for this study was recorded using an IMU containing a triple-axis accelerometer (±16 g) and triple-axis gyroscope (±2000${}^{\circ }/s$). Each player was equipped with two sensors, one for each foot. The sensor is inserted in the insole of the regular football shoe of the player. Due to the stiff material of the insole and the cavity design, the sensor position is fixed without disturbing the players movements. Sensor and insole setup are shown in Figure 1a. Data was recorded with a sampling rate of 200 Hz. For labelling purposes, all session were recorded with at least one video camera.

There were two main requirements for the dataset of our study. First, the amount of data needs to be reasonably high for the training of neural networks. Second, we need real-world data from the pitch for evaluation. As the acquisition of real-world data is complex and time-consuming, we recorded both, laboratory data following controlled protocols and real world data consisting of training and game scenarios. By this dual approach we achieve a high volume of data and are able to compare our algorithms on data with various complexity levels. The acquisitions in the lab contained controlled exercises, e.g., shooting ten penalty kicks and semi-controlled exercises as passing the ball to a team mate, receiving it back and then shooting onto the goal. In the following, we will call those recordings lab sessions. For the real-world acquisitions, hereinafter referred to as field sessions, we are not following a fixed protocol. Instead, football teams were recorded during regular training or games. A typical real-world data recording session is depicted in Figure 1b.

In total, we recorded 181 sessions (38 field sessions/143 lab sessions) with 836 players (97% male/3% female) including youth players starting from German age division U12 to grown up players. The experience of participants ranged from novice players (3%) and amateurs (38%) to semi professional players (50%). For the remaining 9% of players, the skill level was unknown. Due to hardware failure, data of only one foot was recorded in 292 cases. The dataset includes 93,846 labelled ball contacts (8424 shots/24,254 passes/61,168 null). The mean duration of a lab session is 35 min (SD = 17), whereas the mean duration of a field session is 73 min (SD = 38). The mean number of players of a lab session is 4 (SD = 3). For field sessions, the mean number of players is 8 (SD = 4). The maximum number of players in a recording session was 17.

### 3.2. Labelling

The ball contacts were labelled by trained experts. For the classification task, the labels shot, pass and null were annotated. Hereby, a shot was defined as a kick directed towards the goal, a pass as a kick towards a team mate. Any other ball contact, e.g., dribbling, is part of the null class. In real world scenarios we can observe a high variation of the execution of the labelled activities, for example due to differing kicking techniques (e.g., medial versus full instep kick) or distances (long versus short pass). Thus, we used finer grained labelling of ball contacts for further analysis of the classification performance. All ball contact types, a short description and the corresponding label for the classification task are given in Table 1.

### 3.3. Data Preprocessing

For synchronization of IMUs and video data, a characteristic synchronization movement was performed with all sensors of one session simultaneously at the beginning and the end of each data recording session. In addition, the whole session including the synchronization movement was recorded by video. To perform the synchronization movement all sensors were fixed to a rod. Consecutively, the rod was clapped to the ground three times in a row inducing three clearly visible peaks in all accelerometer axes. A CNN model was trained to detect the characteristic synchronization pattern of three consecutive peaks based on the accelerometer signal. Details on the model architecture and model parameters are given in Appendix A. If automatic synchronization failed, the distinctive synchronization pattern was detected manually in the sensor signal. The corresponding moment of the first clap in the video recordings was identified by visual inspection. The time axis of the sensor data was interpolated. For this purpose, the sampling frequency was estimated using the number of samples between the detected synchronization peaks and the recording duration extracted from the videos. The result of the synchronization was manually checked for all recordings by examination of the IMU signal at labelled shot events. Subsequent to synchronization, all IMU signals were scaled between −1 and 1.

### 3.4. Segmentation

First, the data was split into a training and test set. The training set consists of 161 sessions with 697 players, 1171 sensors and 81,165 labelled ball contacts (7362 shots/20,060 passes/53,743 null). For 223 players only data of one foot was recorded due to hardware failure. The test set contains 20 sessions of 139 players, 209 sensors and 12,681 labelled ball contacts with a class distribution comparable to the training set (1062 shots/4194 passes/7425 null class). For 69 players sensor data of only one foot is available because of malfunctions of the sensor. Further, the test set can be divided into 10 sessions acquired in a laboratory setting (lab data) and 10 sessions depicting real world samples recorded during training or games (field data). For training the SVM and neural networks, windows of fixed length were segmented around labelled ball contact events. Banos et al. [17] found that the window size for HAR tasks should be at most 2 s. In Schuldhaus et al. [13], a window size of 1 s was used. Thus, for training the SVM we used 1 s windows. For the neural networks we compared training results for three different window lengths: 1 s, 1.5 s and 2 s. As results were barely differing (see Appendix B), we chose 2 s windows in order to retain information about previous motions.

Figure 2 depicts exemplary IMU data of all three classes. The examples illustrate that ball contacts are clearly visible as peaks in the accelerometer and gyroscope signals. For the shot class example, sensor saturation is reached in the accelerometer data due to the strong impact during ball contact. The last zero crossing of the gyroscope x-axis before ball contact indicates the start of the leg acceleration phase of the kick [39]. For both, shot and pass, the high acceleration of the leg in the sagittal plane is visible. The null class sample shows a dribbling instance. As the distance between adjacent ball contacts can be in the range of milliseconds, a window may contain multiple ball contacts. In the gyroscope data of the null class example window, a gait pattern comparable to those explored in recent work regarding gait analysis as [40] is noticeable. For training, only null class samples containing labelled ball contacts were used. Hence, training did not comprise any general activities as running, jumping or standing. This enables the classifier to learn the split between relevant, hard to distinguish samples as ball contacts during dribbling versus passes.

### 3.5. Adaptation, Optimization and Training of the SVM

For the comparison of the deep learning-based shot and pass classification to literature, the SVM algorithm of [13] was implemented and trained on our dataset. The algorithm comprises a peak detection algorithm, event leg classification, kick phase segmentation and feature extraction step. As we perform the kick and pass detection on both legs separately, the identification of the event leg was neglected in our study. For classification, Schuldhaus et al. [13] implemented a hierarchical architecture. In a first step, null class dribbling instances are identified by peak detection. In the second step, a linear SVM classifier is trained to distinguish shots and passes. Additional ball contact types as dribbling contacts are not present in this processing step. In comparison to Schuldhaus et al. [13], our real-world dataset contains a variety of different ball contact types belonging to the null class. In real-world games we expect dribbling instances with higher speed. Those ball contacts can have a high impact characterized by a peak in the sensor signal similar to kicks. Thus, in our case a peak detection is not sufficient for the recognition of null class samples. As a consequence, we train the SVM algorithm on the three class problem shot versus pass versus null class ball contacts. Kick phase segmentation and feature extraction were performed as described in [13]. Following a biomechanically-driven definition, the kick phase starts with the acceleration of the event leg and ends with the ball contact. To identify the ball contact event, the maximum angular acceleration in the sagittal plane was computed [41]. Afterwards, the last zero-crossing of the angular velocity in the sagittal plane prior to ball contact was determined indicating the start of the leg acceleration [39]. For each accelerometer and gyroscope axis, the absolute sum of the signal during the kick phase from the start of the leg acceleration to the ball contact was computed. The resulting six features were utilized as input of the classifier [13]. Figure 3 shows histograms and scatter plots for two exemplary features. The plot is constructed using shots, medial passes and light ball contacts of two randomly selected lab sessions. The plots indicate separability for samples characterized by equally high values for both features. In contrast, the class affiliation of samples characterized by very low values for both features will be hard to define based on the depicted features. Random under-sampling of the minority classes was used to train on a balanced dataset. For the optimization of the cost parameter *C* of the linear SVM, a 5-fold cross-validation was applied, where each data recording session was either part of the training or validation set. For each fold, a grid search with $C\in \left\{2^{N}\right\},N\in \left\{-10,\ldots ,10\right\}$ was conducted. The model achieving the best weighted mean f1-score over all folds was selected (C = 32) and trained on the whole training dataset.

### 3.6. Deep Learning Architectures

In the field of sensor-based HAR, deep learning methods recently gained a lot of attention. According to Baloch et al. [42], the most common neural network architectures for sensor based activity recognition are CNNs (40%), Recurrent Neural Nets (RNNs) including LSTMs (30%) and hybrid models as convolutional LSTM models (15%) For each of this network types we implemented a basic architecture for the shot and pass classification in football. From the kinetic and kinematic analysis of side-foot and instep kicks in football we know that the two kicking techniques differ in foot speed and foot rotation. In addition, a complex series of rotations is necessary to perform a side-foot kick [39,43]. Thus, we use the segmented event windows including all 3 accelerometer and all 3 gyroscope axes as input to the neural nets. To cope with the data imbalance, we use random under-sampling of the majority classes (null and passes) during training. For hyperparameter optimization, the optuna framework [44] was used applying Bayesian optimization. Unpromising trials were pruned. Within each optimization trial, a 5-fold cross-validation was applied. Samples of one session where either part of the training or validation split. During model training, the batch-wise test f1-score is monitored. The f1-score is an evaluation metric commonly applied for imbalanced data and is computed as the harmonic mean of precision and recall as follows:(1)$$f_{1}=2\;\cdot \;\frac{precision\;\cdot \;recall}{precision+recall}$$

As commonly done for multiclass classification problems in HAR to account for class imbalances, we report the weighted average of the f1-scores of all classes:(2)$${\hat{f}}_{1}=\frac{\sum_{i=1}^{c}w_{i}f_{1}^{i}}{\sum_{i=1}^{c}w_{i}},$$
where *c* depicts the number of classes and $w_{i}$ the number of instances of class *i*. We optimized for 500 trials. The objective function aims to maximize the mean of the weighted f1-scores over all folds. The models were trained for 100 epochs using a batch size of 64. In addition, we used early stopping with patience 40 and reduced the learning rate by a factor of 0.2 if no improvement is seen over the last 20 epochs. The SGD optimizer was used for all models.

#### 3.6.1. 1d Convolutional Neural Net

First proposed in [45] in 1982, CNNs were frequently applied in the field of HAR today due to their scale invariance and their ability to capture local dependencies [46]. A typical CNN architecture consists of one or multiple convolutional layers using learnable filters and nonlinear activation functions such as rectified linear units (ReLUs) for the extraction of increasingly complex features. The manual feature engineering step as known from the development of shallow algorithms is superseded [47]. After each convolution, pooling layers are applied for down sampling of feature maps resulting in local translation invariance [48]. Our proposed CNN consists of three convolutional blocks with ReLU activation functions and max-pooling followed by a fully-connected layer and a softmax [49] output for classification. An overview of all layers and corresponding model parameters is given in Table 2. The search spaces for parameter optimization are summarized in Table 3.

#### 3.6.2. Long Short Term Memory Network

LSTMs are a widely used type of RNNs. RNNs were developed to model sequential data. Due to the vanishing and exploding gradient problem, this property is limited to a short time [34]. To store and output information over longer time periods, LSTMs were developed in 1997 having multiple memory cells with proprietary states [50]. The most popular variant, the vanilla LSTM, consists of three gates, namely the input, forget and output gate regulating to keep or reset the cell state [51]. Thus, general features are identified while preserving temporal dependencies [52]. For LSTMs, the number of trainable parameters is high compared to other deep learning architectures. As a result, the tuning of LSTM parameters can be challenging [53]. We propose an architecture with a single LSTM layer. Batch normalization was applied to improve convergence [54]. A dense layer with soft-max output was used for classification. Thereby, L1 and L2 kernel regularization was utilized. Table 4 shows all layers and the corresponding model parameters of the lstm model. The optimized hyperparameters are summarized in Table 5.

#### 3.6.3. Convolutional Lstm

To combine the advantages of both, convolutional and LSTM layers, hybrid models were developed. The convolutional layers act as feature extractors. The received abstract representation of the input raw signal is consecutively modelled by recurrent layers [55]. This combination outperformed previous results in video-based HAR [56]. Instead of using a time-distributed convolutional layer prior to the LSTM layer, Shi et al. [57] developed the convLSTM, where convolutions are applied in state-to-state and input-to-state transitions. For the shot and pass classification by a convLSTM model, each sample is split into segments. Thereby, the number of segments was optimized within the optuna framework [44]. We added dropout prior to a fully-connected layer. An output soft-max layer was used for classification. All layers and model parameters of the convLSTM model are depicted in Table 6. A summary of all tuned model hyperparameters is given in Table 7.

### 3.7. Evaluation Methods

For model evaluation, the SVM and the optimized neural networks were tested on the test dataset consisting of 10 laboratory and 10 field sessions. Regarding the real world applicability, we evaluated three different scenarios. Thereby, the scenarios differ by testing method (segmented samples versus moving window) and data type (lab data versus field data). Our first evaluation scenario used segmented 2 s windows around labelled ball contacts of the lab test data. Hereafter we will refer to this approach as segmented_lab. In comparison to the first approach, the evaluation of the second and third approach was performed on the whole data stream. For this purpose, a moving window with a length of 2 s and an overlap of 25% was implemented. To reduce the number of windows to be classified, low activity windows are rejected in a previous peak detection step adapted from Schuldhaus et al. [13]. Therefore, the gyroscope signal was processed by a Butterworth high-pass filter with an order of 2 and a cutoff frequency of 20Hz. Then, the signal magnitude vector (smv) of all filtered gyroscope axes is computed and peaks were detected using the peakUtils library [58,59]. The minimum distance between peaks was set to 300 samples (1.5 s) and the threshold was defined as 0.3∗max(smv) with max(smv) being the absolute maximum of the smv in the respective window. Thereby, the Butterworth filter parameters and peak detection parameters were optimized via grid search on 20 randomly selected data recordings from the training dataset by maximizing the specificity with a fixed sensitivity of 100%. For shot and pass candidates, a window of 2 s around the peak was segmented and the raw accelerometer and gyroscope data was used as an input for the classifiers. In order to deal with closely successive ball contacts of the same player, the ground truth label was given by the label of a 1 s window around the detected peak.As the event is located in the center of a segmented window and an overlap of 25% is used, the closest distance of detectable ball contact events is 0.5 s. Within the fastest 5% of consecutive dribbling and pass or shot activities, the mean distance of events is 0.75 s. Smaller distances of around 0.3 s can be observed between dribbling ball contacts. As we do not differentiate dribbling ball contacts from different null class samples, the identification of all ball contacts during dribbling is not necessary. If the label of a window is unobserved, the window was excluded from the evaluation. The described moving window procedure was applied to the lab data for the second evaluation approach (window_lab) and to the field data for the third evaluation approach (window_field).

For the comparison between the SVM performance for full-instep and side-foot kick detection from Schuldhaus et al. [13] and the general shot and pass detection in our approach, the sensitivity of the three classes measuring the proportion of correctly identified samples is computed. The sensitivity of class *c* is given as:(3)$$sensitivity_{c}=\frac{TP_{c}}{TP_{c}+FN_{c}},$$
where the true positives of class *c* ($TP_{c}$) are the number of correctly identified samples belonging to class *c* and the false negatives of class *c* ($FN_{c}$) denote the number of samples erroneously predicted as class *c*. As a performance measure for the comparison of the SVM to the deep learning models we compute the weighted f1-score as given in Equation (Equation 2). To get deeper insights into the confounding factors of the shot and pass detection for lab and field data for the moving window evaluation, we consider the finer grained labels summarized in Table 1. We use an adaptation of the confusion matrix usually showing predictions and true class labels. In our case, we give the finer grained true ball contact along with the true class label. Furthermore, we normalize the columns (true ball contacts) of the confusion matrix. A schematic overview of this visualization is given in Table 8. There are two main reasons for merging different ball contact types into the classes null and pass instead of considering a finer-grained classification problem. First, some ball contact types such as long passes are rare. Thus, a class affiliation based on ball contact types will amplify the imbalance of the problem. Second, the differentiation between different ball contact types was rule-based. A measurable differentiation should be explored in future work. As we did not assess the intra-labeller agreement for ball contact types, the easier distinguishable class labels were used for classification.

## 4. Results

In total, the segmented_lab approach evaluated 3503 labelled ball contacts. For the window_lab evaluation, 104,660 windows were already rejected by the peak detection based candidate selection algorithm. 6883 windows (around 6%) were identified as candidates and further processed by the classification models. No shots or passes were missed by the peak detection algorithm. The identified candidates contained 1777 labelled ball contacts. Roughly half of the labelled null class ball contacts were filtered out by the peak detection algorithm. In the window_field evaluation approach, 38,877 peaks were detected (around 7%), whereas the remaining 501,868 windows were rejected. Event candidates contained 3454 from the 9178 labelled ball contact instances. Again, all labelled shots and passes were identified as candidates by the peak detection.

Table 9 shows the sensitivity for each class achieved by the SVM classifier for our three evaluation approaches. In addition, the results reported by Schuldhaus et al. [13] are given for comparison.

A summary of the performance of the neural nets and the SVM for all evaluation approaches is given in Table 10. The weighted f1-scores for training, segmented_lab, window_lab and window_field are reported. The CNN architecture achieves the highest f1-score throughout all experiments with a peak weighted f1-score of 0.928 in the window_field evaluation. In all experiments, the deep learning approaches outperform the SVM classifier. The training score value depicts the mean weighted f1-score over all cross-validation folds and thus can not be directly compared to the evaluation results. Due to the changed distribution of classes in the moving window scenarios compared to training and the segmented_lab evaluation, the f1-scores of the testing scenarios are frequently higher than the training scores.

To illustrate the classification performance, Figure 4 shows the classification result of all models on an example signal. The 20 s excerpt was recorded during a laboratory session. It includes two passes, a shot and standing as well as running sequences. The peak detection identified 6 event candidates and rejected around 10 s of the sequence. The CNN and convLSTM model could correctly detect both passes and the shot. In contrast, the LSTM missed the second pass and the SVM missed both passes detecting only the shot event.

To understand the influence of different ball contacts on the classification performance, confusion matrices as explained in Section 3.7 for the window_lab and window_field evaluation are depicted in Figure 5 and Figure 6, respectively. In contrast to classical confusion matrices, we subdivide the true classes into labelled ball contacts. For both depicted evaluation methods, the SVM shows a tendency to predict the null class in the majority of windows of all classes. For the deep learning methods, for most instances the highest proportion of a ball contact is predicted to be in its true class. Exceptions are long passes for the LSTM model for both evaluation methods, short passes ambiguous for the LSTM in the window_lab evaluation and shots for the CNN classifier in the window_field evaluation. In general, a small proportion of null class samples is predicted to be a pass but almost no null class instances are confused with a shot. From the different ball contacts belonging to the null class, strong contacts are most likely confounded with a pass. Altogether, the CNN shows superior performance for the classification of shots and passes. For long and ambiguous range passes, especially in the window_field evaluation, convLSTM and LSTM tend to predict a shot instead of a pass. In exchange, they confound less shots with a pass compared to the CNN. In general, the classification performance of shots is significantly smaller for the window_field evaluation than for the window_lab evaluation approach. The worst performance of shot classification in the window_field evaluation is reported for the CNN, where 48% of the shots are correctly identified.

## 5. Discussion

### 5.1. Peak Detection for Candidate Selection

We proposed a candidate selection method adapted from [13] in order to minimize the computational time by reducing the number of null class samples that need to be classified. For both evaluation methods a considerable amount of windows, namely 94% for the evaluation with laboratory data (window_lab) and 93% for the evaluation with real-world data (window_field), could be rejected prior to classification. The remaining windows contain 25.8% and 8.9% labelled ball contacts, respectively. In our study, the main proportion of windows corresponds to unlabelled actions as running or jumping. The classification of those samples is expected to be easy to distinguish from passes and shots as they are not characterized by a ball contact. We adapted the peak detection parameters using the training dataset to assure that, no pass or shot is missed. In practice, it could be intentional to reject passes and shots with very low impact assuming they indicate unintentional activities. The peak detection parameters can easily be adapted accordingly. Even though a more sophisticated candidate selection could be developed, the peak detection worked very well for shot and pass detection. In [8], a similar approach was successfully used for impact detection in beach volleyball indicating the applicability of the proposed method for disciplines as tennis, badminton or squash, where the sport-specific movements are evenly characterized by high impact events.

### 5.2. Svm Performance from Laboratory to Real-World Scenarios

In order to compare the performance of a common shallow classifier established for laboratory settings, we adapted the SVM from Schuldhaus et al. [13]. In the original publication, side-foot kicks and full-instep kicks were classified. The model was trained on segmented data from controlled exercises recorded in a laboratory setting. A hierarchical approach was implemented. All null class samples were rejected via peak detection. Thus, a two-class classification problem was evaluated. The SVM achieved high sensitivities of 97.9% and 95.6% for side-foot kicks and full-instep kicks, respectively [13]. We assumed that the majority of passes in real-world scenarios resembles the side-foot kicks, whereas shots are commonly carried out with the instep. Hence, we used the same features for classification of shots and passes in general. As we included a bigger variety of null class samples, peak detection was not sufficient to reject all null class instances. In consequence, a more complex three class problem needs to be solved. To ensure comparability to [13], in the segmented_lab evaluation we evaluate on segmented samples of laboratory data. Whereas a high sensitivity of 93.3% is reported for the null class, very poor sensitivity is observed for shots and passes (38.8% and 21.7%, respectively). Similar results are reported for the window_lab and window_field evaluation approaches with slightly worse performance for window_field. The adaptations of the confusion matrices for window_lab and window_field show that despite the training with balanced class distributions, the SVM predicts a null class sample over proportionally often. Even for the short pass medial, the ball contact label closest to side-foot kicks with regards to technique and contact point of ball and foot, only 26% of samples in case of window_lab and 21% of samples in case of window_field are correctly identified as pass. The chosen features are not discriminative for the given three-class problem, highlighting the importance of domain-knowledge for feature-based approaches. In the scope of this study we did not assess discriminatory power of different features. The applicability on real-world data for the original two-class problem is not evaluated in the present study.

### 5.3. Deep Learning-Based Classification in Real-World Scenarios

In the following paragraph, the potential of different neural network types for real-world activity recognition in sports is discussed. In comparison to the SVM, the achieved f1-scores show the general feasibility of the classification task. Overall, the best performance is reported for the CNN architecture. Uncommonly, the reported f1-scores for all three evaluation scenarios are higher than for the model training. This can be explained by the changed distribution of null class samples, especially in the window_lab and window_field case, where additional unlabelled null class samples are included. Moreover, the weighted f1-score takes class imbalances into account assigning a higher weight to the null class samples. Thus, it is important to examine the results from Figure 5 and Figure 6 to gain insights into the classification performance for the shot and pass classes. The overall distribution confirms the general ability of all deep learning models to correctly identify shots and passes. Thereby, the strength of the impact caused by the ball contact seems to play an important role. This would explain the low number of shot predictions for null class samples and the difficulty to correctly assign long passes to the pass class while still achieving a high sensitivity for the shots.

We inspected the corresponding video footage for a randomly selected subset of wrongly classified instances. We observed that a high proportion of the underlying actions of misclassified null class samples can be described as agile containing sudden changes of direction or speed, defensive cutting movements or tackling situations with contact to another player. Those movements are expected to be more common in real-world scenarios than controlled exercises. For shots we can observe a significant drop in sensitivity between the window_lab and and window_field evaluation from 75% to only 45% for the CNN architecture. The manual inspection of the videos showed that especially short distance shots, low intensity shots, medial kicks directed towards the goal and volleys are prune to classification errors. We conclude that such instances are more likely to appear in the uncontrolled environment of real-world acquisitions. Future studies should investigate the influence of agile movements without ball contacts in greater detail. In this context, sensor information with regards to upper body movement is particularly interesting. Furthermore, the deep learning-based models are incapable of identifying long passes as pass and simultaneously distinguishing low intensity shots from passes reliably. The division of long and short passes during labelling was not based on a specific distance, but rather motivated by different kicking techniques. Still, a clear division of classes was difficult. Measuring the labelling agreement between multiple annotators gives valuable insights, but was outside the scope of this study.

All experiments conducted in this study were performed off-line. However, real-time performance assessment can be important in training and game scenarios. In this work, we showed that the integration of a peak detection-based candidate selection can be successfully applied to further reduce computational complexity. Furthermore, we limited the depth of the models to keep computational effort manageable. For the CNN, 3 convolutional layers were used. The LSTM and convLSTM architecture consist of a single LSTM or convLSTM layer. Comparable architectures were already adopted on mobile devices and used for real-time predictions indicating the feasibility of executing our trained models on wearable devices [35,60]. However, further work is needed to enable incremental online learning [61]. Thereby, the challenge of unsupervised HAR including the recognition of unseen activities needs to be addressed in future work [62].

### 5.4. General Aspects for Activity Recognition in Real-World Scenarios

In literature, the general feasibility of sensor-based activity recognition in the field of sports was shown in various areas for laboratory environments. However, our study indicates that the shift to real-world scenarios is not trivial and needs to be performed carefully. A slight change of the task definition adapted to real-world needs can lead to undiscriminating features. Due to high variations in technique, intensity, distances and the influence of opponents the task gains additional complexity. This already played a role in laboratory environments when kicking technique was not specified for otherwise controlled exercises. In our study, different age groups from kids over adolescents to grown ups were considered. Only a small number of female players are included. The influence of age, weight, height, gender or skills on classification performance should be investigated in future work. Moreover, the personalization of the models for age groups or even individuals is an interesting direction for future work. In contrast to existing studies, we used a large database for the training of the deep neural networks. This enables the exploration of transfer learning in the field of activity recognition in sports in future work. By transfer learning techniques, models trained in one domain can be fine-tuned to operate on data of similar domains leading to high classification accuracies while simultaneously reducing training time. In other application areas, transfer learning for activity recognition is already a vivid field of research [62]. In addition, it was recently applied to develop personalized HAR models [12].

## 6. Conclusions

In this study we investigated the potential of an SVM model from literature established in a laboratory setting for the detection of shots and passes under real-world conditions. Instead of the published full-instep and side-kick classification, we aimed to identify shots and passes regardless their kicking technique. Moreover, we developed different types of deep learning models, namely a CNN, a convLSTM and a LSTM model to explore their potential for shot and pass detection in real-world conditions. For this purpose, we recorded IMU data from over 800 football players in laboratory and real-world settings. For model evaluation, we consider three different scenarios to model different stages of complexity. We evaluated the models on segmented samples of ball contact instances from laboratory data (segmented_lab). For the second and third approach, we introduced a sliding window approach and adapted a peak detection method from literature as candidate selection prior to classification. This was applied on laboratory data (window_lab) and real-world data (window_field). The results demonstrate that the shallow SVM architecture is not able to distinguish passes and shots in all three evaluation scenarios. This shows that the features are not discriminative for the classification of shots, passes and other ball contacts. All deep learning models outperformed the SVM model and show the feasibility of real-world activity recognition in sports for tasks of practical relevance. Thereby, the best performance could be achieved by the CNN model with a mean f1-score of 92.8% in training and game scenarios. Our study reveals important considerations for the assessment of existing laboratory studies regarding their transferability to real world scenarios. In particular, it highlights the importance of reevaluating the discriminative power of shallow features developed under laboratory conditions for real-world activity recognition. It shows that scientists who aim to set up studies with relevance for real-world applications should carefully select an appropriate dataset and evaluation approach. Moreover, we illustrated that in contrast to feature-based methods, deep neural nets can learn high-level representations of complex activities with variations in execution. Thus, deep-learning-based methods for real-world HAR in sports should be explored in more detail in future work.

## Figures and Tables

**Figure 1 sensors-21-03071-f001:**
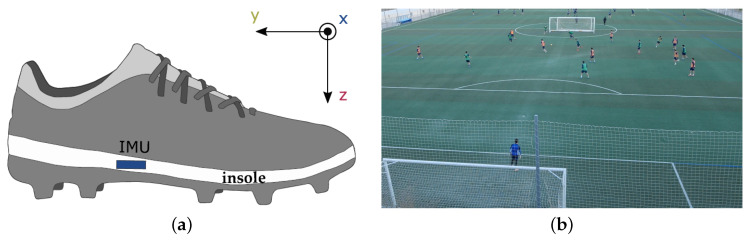
Hardware set up and example of in-field data acquisition. (**a**) Schematic representation of the position of the IMU sensor in the insole of a conventional football boot and the sensor coordinate system. (**b**) Example of a typical real-world data recording during a regular training session. The image shows the camera view used for labelling.

**Figure 2 sensors-21-03071-f002:**
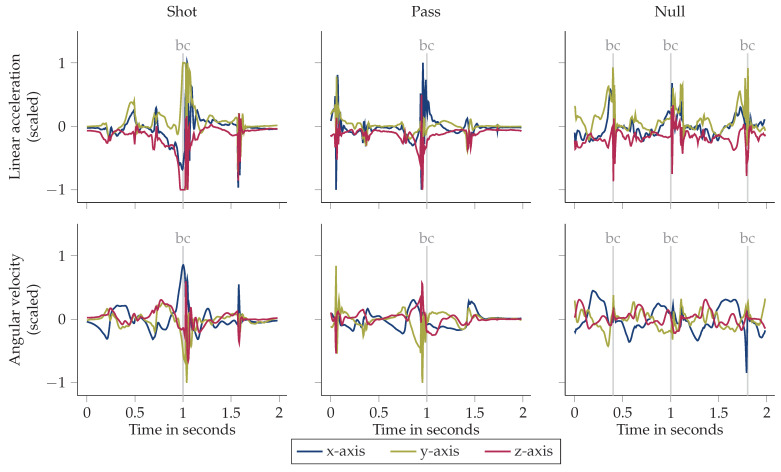
Exemplary IMU data for each class. Each ball contact (bc) is indicated by a vertical gray line. Sensor orientation is shown in Figure 1a.

**Figure 3 sensors-21-03071-f003:**
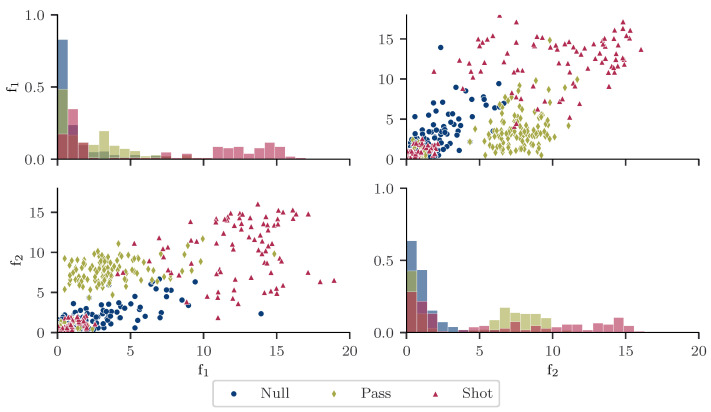
Histograms and scatter plots for two exemplary features: The absolute sum of the accelerometer x-axis during the kick phase (f${}_{1}$) and the absolute sum of the accelerometer y-axis during the kick phase (f${}_{2}$). Only shots, medial passes and light ball contacts from 2 random laboratory training sessions are considered. The histograms are constructed with 25 bins and show frequency density. In addition, the sum of the bars of each histogram is normalized and bars for different classes are stacked.

**Figure 4 sensors-21-03071-f004:**
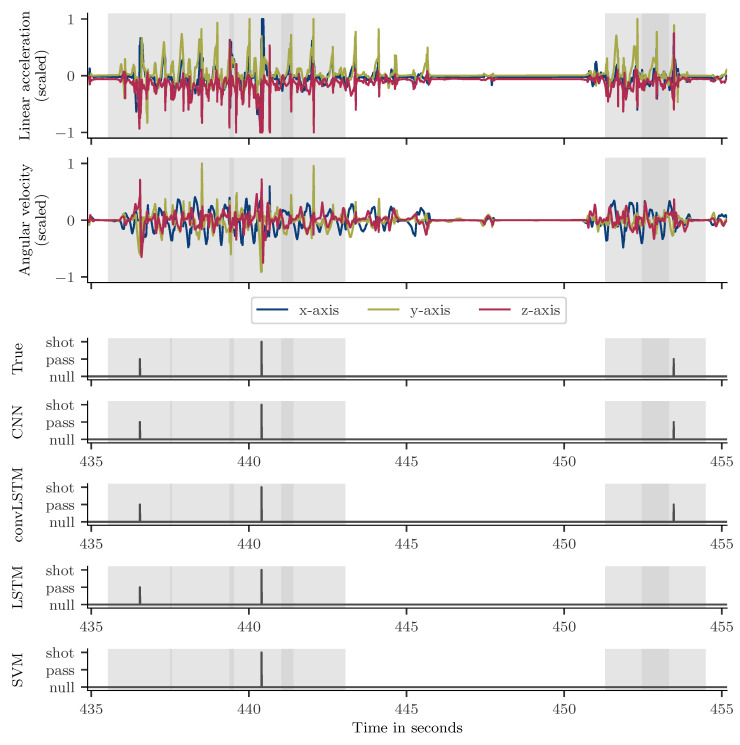
Exemplary accelerometer and gyroscope data excerpt from a laboratory session. The upper two plots show the accelerometer and gyroscope data. Below, the true class label and the predictions by the respective model are shown. At the beginning of the exercise sequence, the player is standing. He passes the ball (second 436) and sprints towards the goal. After receiving the ball back from another player (second 439), he dribbles towards the goal and shots (second 440.5). He runs to a cone, where he is standing until second 451. Then, he sprints towards the ball and passes it to another player at second 453. The peak detection algorithm identified 6 event candidates.

**Figure 5 sensors-21-03071-f005:**
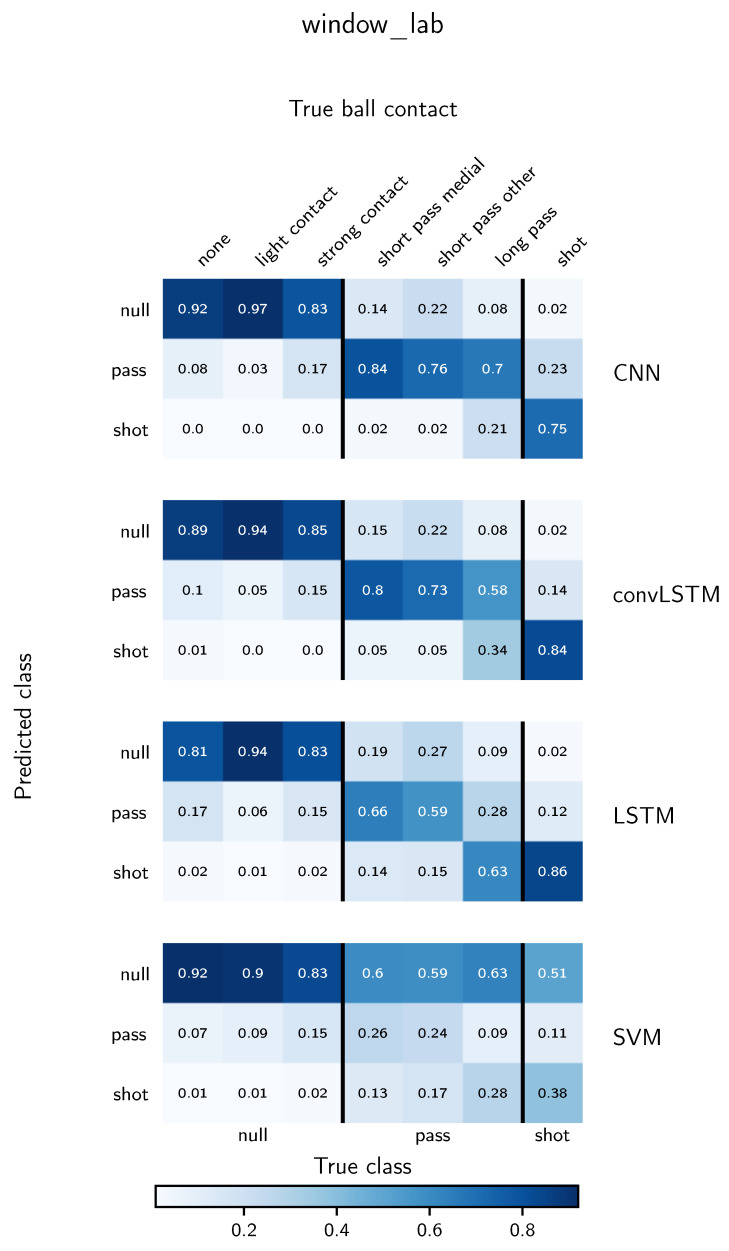
Classification performance of all models for the window_lab evaluation using a moving window and candidate selection on laboratory data. A detailed explanation of the way of presentation can be found in Section 3.7. True and predicted classes are given on the bottom and left, respectively. Finer grained true ball contacts are shown on top. The given values are normalized over the true ball contact labels. Thus, the greatest possible value is one, the smallest possible value is zero.

**Figure 6 sensors-21-03071-f006:**
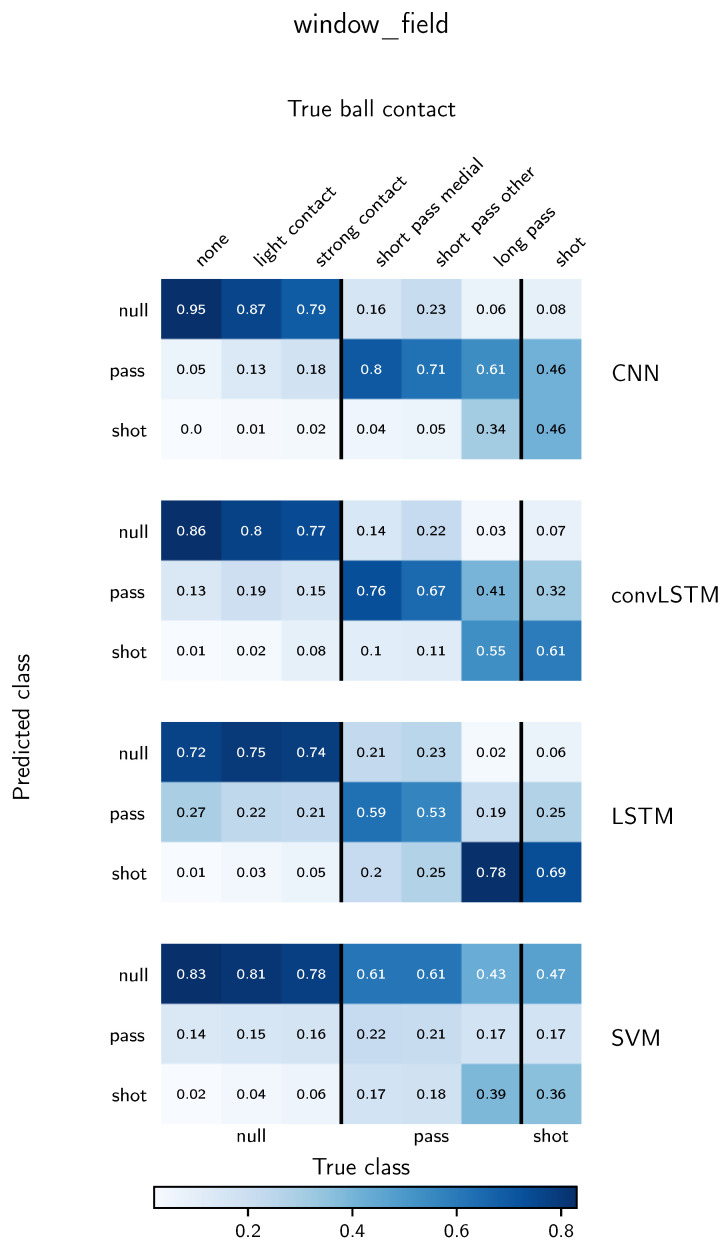
Classification performance of all models for the window_field evaluation using a moving window and candidate selection on laboratory data. A detailed explanation of the way of presentation can be found in Section 3.7. True and predicted classes are given on the bottom and left, respectively. Finer grained true ball contacts are shown on top. The given values are normalized over the true ball contact labels resulting in the greatest possible value being one and the smallest possible value being zero.

**Table 1 sensors-21-03071-t001:** Overview of the ball contact types, their definitions and their class affiliation for the three-class problem shot versus pass versus null.

Ball Contact Type	Description	Class
none	no ball contact, player is for example running or sprinting	null
light contact	small contact while player is not moving or during dribbling	null
strong contact	strong ball contact, e.g., bringing the ball forward during dribbling	null
short pass medial	short pass with the medial part of the foot	pass
short pass other	pass with another part of the foot than medial	pass
long pass	pass over longer distance	pass
shot	kick directed towards the goal	shot
unknown	player not visible in video	excluded

**Table 2 sensors-21-03071-t002:** Overview of all layers and model parameters of the proposed CNN architecture. The output shape is given for one batch.

Layer Type	Hyperparameter	Output Shape	# of Parameters
1D Convolution	filter${}_{1}$: 256, kernelsize: 3	(398, 256)	4864
Max-pooling	poolsize: 2	(199, 256)	0
1D Convolution	filter${}_{2}$: 128, kernelsize: 3	(197, 128)	98,432
Dropout	dropout${}_{1}$: 0.30228	(197, 128)	0
Max-pooling	poolsize: 2	(98, 128)	0
1D Convolution	filter${}_{3}$: 16, kernelsize: 3	(96, 16)	6160
Dropout	dropout${}_{2}$: 0.03576	(96, 16)	0
Max-pooling	poolsize: 2	(48, 16)	0
Fully-connected		(768)	0
Dropout	dropout${}_{3}$: 0.43372	(768)	0
Dense		(3)	2307

**Table 3 sensors-21-03071-t003:** Hyperparameter, search space and optimization results for the CNN model.

Hyperparameter	Sampling	Search Space	Result
filter${}_{1}$	categorical	$2^{N},N\in \left\{6,\ldots ,9\right\}$	256
filter${}_{2}$	categorical	$2^{N},N\in \left\{4,\ldots ,8\right\}$	128
filter${}_{3}$	categorical	$2^{N},N\in \left\{4,\ldots ,8\right\}$	16
poolsize	categorical	∈{2,3,4,5}	2
kernelsize	categorical	∈{2,3,4}	3
dropout${}_{1}$	log uniform distribution	∈[0.01,1)	0.30228
dropout${}_{2}$	log uniform distribution	∈[0.01,1)	0.03577
dropout${}_{3}$	log uniform distribution	∈[0.01,1)	0.43372
learning rate	log uniform distribution	∈[0.001,0.1)	0.09996

**Table 4 sensors-21-03071-t004:** Overview and details of the LSTM architecture.The output shape is given for one batch.

Layer Type	Hyperparameter	Output Shape	# of Parameters
LSTM	units: 64	(64)	18,432
BatchNorm		(64)	256
Dense		(3)	195

**Table 5 sensors-21-03071-t005:** Hyperparameters, search space and optimization results for the LSTM model.

Hyperparameter	Sampling	Search Space	Result
units	categorical	$2^{N},N\in \left\{6,\ldots ,9\right\}$	64
L1 regularizer	log uniform distribution	∈[0.0001,0.1)	0.00013
L2 regularizer	log uniform distribution	∈[0.0001,0.1)	0.00111
clipvalue	log uniform distribution	∈[0.1,0.8)	0.19517
learning rate	log uniform distribution	∈[0.001,0.1)	0.06955

**Table 6 sensors-21-03071-t006:** Overview and details of the convLSTM architecture.The output shape is given for one batch.

Layer Type	Hyperparameter	Output Shape	# of Parameters
convLSTM2D	filter${}_{1}$: 128, kernelsize: 3	(1, 48, 128)	206,336
Dropout	dropout: 0.22620	(1, 48, 128)	0
Fully-connected		(6144)	0
Dense		(3)	18,435

**Table 7 sensors-21-03071-t007:** Model parameters and hyperparameters, search space and optimization results for the convLSTM model.

Hyperparameter	Sampling	Search Space	Result
filter	categorical	$2^{N},N\in \left\{4,\ldots ,8\right\}$	128
kernelsize	categorical	∈{2,3,4}	3
dropout	log uniform distribution	∈[0.01,1)	0.22620
num segments	categorical	∈{2,4,5,8,10}	8
learning rate	log uniform distribution	∈[0.001,0.1)	0.09403

**Table 8 sensors-21-03071-t008:** Example of our adaptation of the normalized confusion matrix to show classification performance for detailed ball contact labels (bcs). $P_{c,b}$ denotes predictions of class *c* of instances of the true ball contact *b*. For orientation, we give the true class on the bottom. Light color represents low values, dark color represents high values. For an optimal classifier $P_{A,1}$, $P_{A,2}$, $P_{B,3}$ and $P_{C,4}$ are one and all other cells contain zeros.

		True Ball Contact			
Prediction		bc 1	bc 2	bc 3	bc 4
	class A	$P_{A,1}$	$P_{A,2}$	$P_{A,3}$	$P_{A,4}$
	class B	$P_{B,1}$	$P_{B,2}$	$P_{B,3}$	$P_{B,4}$
	class C	$P_{C,1}$	$P_{C,2}$	$P_{C,3}$	$P_{C,4}$
		class A		class B	class c
		True class			

**Table 9 sensors-21-03071-t009:** Comparison of the classification result from Schuldhaus et al. [13] and our own evaluation approaches segmented_lab, window_lab and window_field for the SVM classifier. Schuldhaus et al. [13] classified full-instep versus side-foot kicks (marked with *) on segmented laboratory data, whereas our approaches classify shots and passes in general. In segmented_lab, segmented events extracted from lab data are used for evaluation. window_lab and window_field apply moving windows, peak detection for candidate selection and classification to laboratory and field data, respectively. In accordance with Schuldhaus et al. [13] we reported the class specific sensitivity scores.

Class Label	Schuldhaus et al. [13]	Segmented_Low	Window_Low	Window_High
Null	-	93.3%	91.7%	83.2%
Pass (Side-Foot *)	97.9%	21.7%	23.5%	20.9%
Shot (Full-Instep *)	95.6%	38.8%	37.9%	36.0%

**Table 10 sensors-21-03071-t010:** Results of the shot and pass classification. The table reports the weighted f1-score for all architectures for training and three evaluation methods. segmented_lab uses segmented windows of laboratory data for evaluation. In window_lab and window_field, a moving window is used for segmentation. For each window a peak detection algorithm identifies event candidates. Only identified event candidates are used for the consecutive classification of shots and passes. window_lab uses low complexity laboratory data, whereas window_field evaluated on high complexity data from training and competition. The best f1-score of each experiment is highlighted in bold.

	Training	Segmented_Lab	Window_Lab	Window_Field
SVM	0.648	0.656	0.807	0.815
CNN	**0.887**	**0.923**	**0.912**	**0.928**
LSTM	0.830	0.890	0.840	0.777
convLSTM	0.857	0.910	0.897	0.869

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
