# Peer review of "From the Laboratory to the Field: IMU-Based Shot and Pass Detection in Football Training and Game Scenarios Using Deep Learning"

_sensors, 2021, doi:10.3390/s21093071_

Round 1

Reviewer 1 Report

The paper presents interesting experiments and results. However, I have some concerns that need to be resolved. 

  1. On page 2 line 38-39, the authors claimed "Results of recently developed deep learning models showed superior performance over conventional machine learning methods as Support Vector Machines (SVMs) in vision-based human activity recognition (HAR) [6]". It is inappropriate to mention that deep learning is superior to machine learning in vision-based HAR since this paper focuses on sensor-based HAR. It is highly recommended to discuss the superior performance of deep learning in sensor-based HAR. The following reference gives a comprehensive review of deep learning on sensor-based HAR:

Chen, Kaixuan, et al. "Deep learning for sensor-based human activity recognition: overview, challenges and opportunities." arXiv preprint arXiv:2001.07416 (2020).

Sensor-based and vision-based human activity recognition: A comprehensive survey. https://doi.org/10.1016/j.patcog.2020.107561

2. In section 3.4, how 161 sessions come from 697 players and with 1,171 sensors? How many players in one lab session and field session respectively? In my understanding, each player was equipped with 2 IMU sensors, so 697 players should have 1,394 sensors instead of 1,171. Please provide and clarify more details.

3. How long is a session? How long is a ball contact? How do you deal with transition periods? Please provide more details.

4. Although this paper proposed a real-world experiment, this is still off-line. Could you do on-line experiments? If not, what are the challenges? Please have a discussion on on-line experiments,

5. Could you give a visualization on sensor signals versus classification results for an intuitive understanding your results? 

Reviewer 2 Report

The paper is of good quality and discusses an important topic. However, some changes are required. Please address the comments below:

  1. At the end of the abstract, the authors mention that: “The study provides valuable insights for sports assessment under practically relevant conditions” what are these valuable insights? be more specific.
  2. Page 3, line 109, the abbreviation RF is used without definition. Please define this abbreviation.
  3. To give more insight it is important to illustrate the acceleration and gyroscope data for different classes (pass, shot, and null). This will allow the reader to see how the pattern of the IMU data varies as the ball contact type changes.
  4. On page 6, lines 218 and 219, the authors mention that synchronization is automatically done using a CNN model. Please elaborate more on this and specify the structure of the CNN model and explain how the CNN model is able to do the synchronization. An illustration with data would be helpful to gain a better understanding. Do the authors mean that the IMU data is synchronized with the video data?
  5. For the input signal, we have 6 streams: 3 accelerations signals along 3 axes and angular velocity along three axes. It is not clear if the input for the deep learning algorithm is one of these 6 input signals or what exactly. Please clarify this point and address it in more details and justify your choice
  6. In table 2, page 8, for the output shape the number 64 is not clear. Do the authors mean that the input signal is composed of 64 streams each comprising 400 samples. What are these 64 streams exactly?
  7. For SVM Provide the list of features that are extracted. Also, indicate from which input signal each feature is extracted?
  8. Illustrate using histogram for some of the extracted features for SVM allows differentiating various types of ball activities (shot, pass, and null).
  9. For table 8 on page 11, please justify why you do not split class A into two classes based on the ball contact type instead of including two types of ball contact into a single class A.

Round 2

Reviewer 1 Report

The authors have successfully addressed my concerns. I suggest acceptance in the current format.